

# Subseasonal precipitation forecasts of opportunity over southwest Asia

Melissa Leah Breeden[1,2], John Robert Albers[1,2], Andrew Hoell[2]

[1]Cooperative Institute for Research in Environmental Sciences, University of Colorado Boulder, Boulder, CO
[2] NOAA Physical Sciences Laboratory, Boulder, CO
*Correspondence to*: Melissa L. Breeden (melissa.breeden@noaa.gov)

**Abstract.** Subseasonal forecasts of opportunity (SFOs) for precipitation over southwest Asia during January-March at lead times of 3-6 weeks are identified using elevated expected forecast skill from a Linear Inverse Model (LIM), an empirical
dynamical model that uses statistical relationships to infer the predictable dynamics of a system. The expected forecast skill from this LIM, which is based on the atmospheric circulation, tropical outgoing longwave radiation and sea surface temperatures, captures the predictability associated with many relevant signals as opposed to just one. Two modes of variability, El Niño-Southern Oscillation (ENSO) and the Madden Julian Oscillation (MJO), which themselves are predictable because of their slow variations, are related to southwest Asia precipitation SFOs. Strong El Nino events, as observed in 1983,
1998, and 2016, significantly increase the likelihood by up to threefold of an SFO 3-4 and 5-6 weeks in advance. Strong La Nina events, as observed in 1989, 1999, 2000, also significantly increased the likelihood of an SFO at those same lead times. High amplitude MJO events in phases 2-4 and 6-8 of greater than one standardized departure also significantly increases the likelihood of an SFO 3-4 weeks in advance. Predictable atmospheric circulation patterns preceding anomalously wet periods indicate a role for anomalous tropical convection in the SPCZ region, while suppressed convection is observed preceding
predictable dry periods. Anomalous heating in this region is found to distinguish wet and dry periods during both El Niño and La Niña conditions, although the atmospheric circulation response to the heating differs between each ENSO phase.

## 1 Introduction

Precipitation over Southwest Asia, defined here as 50-80°E, 22-48°N, occurs mainly during the cold season from November – April and determines the regions subsequent water supply used for agriculture and consumption (Agrawala et al. 2001;
Barlow et al. 2006). It is in this context that accurate precipitation predictions are critical, given the effect of water on food security and livelihoods (Famine Early Warning Systems Network, 2022) in this semi-arid region (Barlow et al. 2016). However, precipitation forecasts from dynamical models lack skill by lead times of three weeks over Southwest Asia (de Andrade 2018; Pegion et al. 2019). An alternative endeavor is identifying the smaller subset of forecasts at lead times of three to six weeks that *are* skillful, so-called 'subseasonal forecasts of opportunity' (SFOs; Lang et al. 2020; Mariotti et al. 2020).
SFOs can arise from slowly evolving tropical phenomena such as the El Niño-Southern Oscillation (ENSO; Newman et al. 2003; Johnson et al. 2014; Domeisen et al. 2019; Mariotti et al. 2020; Albers and Newman 2021), and Madden-Julian



Oscillation (MJO; Rodney et al. 2013; Johnson et al. 2014; Li and Robertson 2015; Mayer and Barnes 2021), which can force atmospheric circulation patterns to the extratropics through anomalous divergence (e.g., Sardeshmukh and Hoskins 1988).

Precipitation over southwest Asia may be a favorable target variable and location for SFOs, as both ENSO (Barlow et al. 2002; Nazemosodat and Ghasemi 2004; Hoell et al. 2012; Hoell et al. 2014a; Hoell et al. 2014b; Hoell et al. 2015a; Hoell et al. 2015b; Hoell et al. 2017; Hoell et al. 2018a) and the MJO (Barlow et al. 2005; Nazemosodat and Ghaedamini 2010; Hoell et al. 2013; Cannon et al. 2017; Hoell et al. 2018b) can modulate precipitation in this region. Hoell et al. (2018a) considered the sensitivity of southwest Asian precipitation to central Pacific (CP) and eastern Pacific (EP) El Niño and La Niña conditions

and found that while both CP and EP El Niño conditions shifted precipitation anomalies towards the upper tercile, a wide range of extreme precipitation outcomes was still possible. CP La Niña events shifted precipitation towards the lower tercile, while EP La Niña events did not significantly shift precipitation in the region. Hoell et al. (2018b) found that, in the five days following MJO phases 2-4 (enhanced eastern Indian ocean convection), negative precipitation anomalies developed as a response to an anomalous upper-level anticyclone and subsidence. Conversely, MJO phases 6-8 (suppressed eastern Indian

ocean convection) were associated with anomalous upper level troughing, ascent, and positive precipitation anomalies. However, Cannon et al. (2017) found a nuanced impact of the MJO on extreme western Himalayan snowfall, due to competing influences of the MJO on the dynamic forcing for vertical motion and moisture availability. The convolved impact of ENSO and MJO activity on precipitation in the region is cited as an additional confounding factor that can obscure the nature of the two teleconnections (Schrage et al. 1999; Hoell et al. 2013; Riddle et al. 2013). Objective methods for considering this

combined influence, and how it leads to SFOs, are therefore of interest to both better provide real-time subseasonal forecast guidance and to understand sources of predictability.

Based on past success, this study uses a linear inverse model (LIM; Penland and Sardeshmukh 1995), and its associated signal-to-noise metric 'expected skill' (Sardeshmukh 2000; Newman et al. 2003) to anticipate SFOs over Southwest Asia. Albers and

Newman (2019) used expected skill to identify SFOs, at the time of forecast, for North Pacific and North Atlantic 500-hPa geopotential height anomalies. They found that periods of elevated expected skill forecast by the LIM identified more skillful forecasts compared to the ECMWF IFS and National Center for Environmental Prediction Climate Forecast System Version 2 (NCEP CFSv2) initialized forecast systems, implying that sources of predictability are common among various model types. Albers and Newman (2021) found that NAO SFOs could be identified in both the LIM and IFS, and were driven by a set of

ENSO-related climate modes, reflecting the utility of the LIM in targeting regional phenomena.

Global processes and their unique interactions with local precipitation and temperature ultimately produce SFOs and can be identified by training a LIM on specific regional/large-scale interactions, as demonstrated in Breeden et al. (2022) for North American 2-meter temperature (2mT). Based on these results, here we develop a LIM for subseasonal precipitation over

southwest Asia that has been designed in a similar manner. We will show that precipitation SFOs determined using LIM



expected skill can successfully be identified for subseasonal precipitation over southwest Asia with a LIM that is regional in precipitation and temperature but large-scale with the inclusion of hemispheric tropical outgoing longwave radiation (OLR) and sea surface temperatures (SSTs) and upper-level Northern Hemisphere streamfunction. Another beneficial quality of the LIM is the negligible computational power needed to generate a long record of hindcasts.


The LIM developed for regional precipitation over southwest Asia is used to test the hypothesis that SFOs can be anticipated using theoretical expected skill and are associated with strong ENSO and MJO events. Section 2 introduces the reanalysis and satellite products employed, how the LIM is constructed, and how SFOs are identified using expected skill. Section 3 shows a comparison of this approach to other methods of anticipating periods of elevated forecast skill, and the correspondence between

forecasts of opportunity and ENSO and the MJO. Section 4 contextualizes results and proposes next future steps.

## 2 Data and Methods

### 2.1 Data

To train the LIM, 200-hPa streamfunction ($\Psi_{200}$), 2-meter temperature (2mT), sea surface temperature (SST), and outgoing longwave radiation (OLR) data from Japanese Meteorological Agency 55-year Reanalysis (JRA-55; Kobayashi et al. 2015)

and precipitation from the Climate Hazards InfraRed Precipitation with Stations (Funk et al. 2015) dataset are used for the period January - March, 1981-2020. Variables and their respective domains are listed in Table 1. To consider how forecast skill and forecasts of opportunity change as a function of ENSO phase, the Niño3.4 index was calculated using SST anomalies averaged from 5°N-5°S, 170°W-120°W. For examining relationships between skill and the MJO, the real-time multivariate MJO (RMM) index, a combined tropical OLR and circulation index that is designed to capture characteristics of the MJO

(Wheeler and Hendon 2004), is employed, including RMM amplitude, which measures MJO strength, and each day's associated MJO phase, which tracks MJO location. Finally, based on results in Section 3b, we assess the strength of tropical OLR anomalies in the south Pacific Convergence Zone (SPCZ; box in Fig. 9), defined as 10°S-2.5°N, 140-180°E. The time series of OLR anomalies in this region is considered as a third metric, in addition to Niño3.4 and RMM, that might increase the likelihood of an SFO occurring.

### 2.2 Linear Inverse Model

A LIM assumes that the evolution of a subset of climate anomalies, defined in the state vector **x**, can be approximated as the sum of a slowly evolving, potentially predictable component and a rapidly-decorrelating, unpredictable component. Here we consider the evolution of the following climate variables (Eq. (1-2); Table 1):

$$\mathbf{x} = \{SST, OLR, \Psi_{200}, 2mT, Precip\} \qquad \{1\}$$



$$\frac{d\mathbf{x}}{dt} = \mathbf{L}\mathbf{x} + \mathbf{F_s} \qquad \{2\} \qquad ,$$

where the dynamic operator **L** represents the predictable component of the evolution, and $\mathbf{F_s}$ represents state-independent white
noise forcing that is unpredictable. The LIM employed in this study is created using **L**, designed to capture 'slow and predictable' weekly-timescale variability, as in past studies (Winkler et al. 2001; Newman et al. 2003; Albers and Newman 2019; Breeden et al. 2020). Any predictable processes represented by the variables in **x** are aggregated in the operator **L**, and their net effect on the predictable evolution of the system is leveraged in the LIM (i.e., ENSO/MJO variability). To the extent the key predictable relationships between the LIM variables, whether truly linear or nonlinear, can be estimated linearly
through the covariance between model variables (Eq. (3)), they can be represented by **L**. As such, the LIM can include more information than what is retained in models based on the linearized equations of motion that explicitly exclude nonlinear effects.

Consistent with focusing on the predictable, weekly-varying component of the system evolution, the climate anomalies used
in the LIM were calculated by removing the 40-year daily climatology and then applying a 7-day running mean. Many variable combinations and regional domains were tested, and the combination used here was found to produce the highest precipitation forecast skill during SFOs over southwest Asia. Since the leading EOFs retained in **x** are not sensitive to small changes in the 2mT and Precip domains selected, LIM forecasts and SFOs are also not sensitive to small changes in the regional domains. Note that information contained in variables that are not included in this LIM, for instance slowly evolving soil moisture, can
still be implicitly included in the LIM variables, such as 2mT.

The instantaneous $\mathbb{C}_\mathbf{0}$ and 5-day lagged covariance $\mathbb{C}_{\tau_0}$ between the state vector components are used to determine **L**:

$$\mathbf{L} = ln(\mathbb{C}_{\tau_0} * inv(\mathbb{C}_\mathbf{0}))/\tau_0 \qquad \{3\}$$


As a practical consideration to reduce the dimensionality of **L**, each variable in **x** is truncated using empirical orthogonal function (EOF) analysis, where enough EOFs are retained to capture most of the variance in each variable and region (Table 1). The covariance and lagged covariance are then computed using the principal components that are retained.

A specific training lag $\tau_0$ must be selected to compute the lagged covariance. If the system were perfectly linear and forced by white noise, **L** would not be sensitive to the training lag, but in practice there are constraints to the range of training lags that are appropriate, which is determined using the 'tau-test' (Penland and Sardeshmukh 1995). For this LIM, a training lag of five days is used, which is consistent with the range of stable training lags for LIMs similar to the one used here (Winkler et al.



2001; Newman et al. 2003; Breeden et al. 2020; Henderson et al. 2020). For further information on the sensitivity of weekly
LIMs to training lag and additional parameters, the reader is referred to Section 5 of Winkler et al. 2001.

Daily LIM forecasts are generated using initial conditions $\mathbf{x}(0)$ for any lead time, $\tau$, by solving the homogeneous component
of Eq. (2) (Penland and Sardeshmukh 1995):

$$\hat{\mathbf{x}}(\tau) = \mathbf{x}(0)\exp(\mathbf{L}\tau) = \mathbf{x}(0)\mathbf{G}(\tau) \quad \{4\}$$

LIM forecast skill is assessed using 10-fold cross-validation, done by removing 10% of the data, re-computing $\mathbf{L}$, and
generating forecasts for the 10% of initializations that were removed (e.g., Albers and Newman 2019). This process is repeated
to generate forecasts for 1981-2020, initialized on 1 January – 20 March each year. LIM forecast skill is assessed two ways:
using anomaly correlation coefficient (ACC), where the LIM forecast and un-truncated (i.e., full-field) verification
precipitation is compared at each grid point, and using the pattern correlation coefficient (PCC), where the LIM forecast and
un-truncated verification precipitation are compared at each timestep using the uncentered, cosine-weighted correlation
between all grid points over southwest Asia (e.g., Albers and Newman 2019). ACC measures skill over the full period, while
preserving geographic information about skill, while PCC measures the average skill over the entire southwest Asian domain
but maintains information about how individual forecasts performed.

### 2.2.1 Forecasts of Opportunity

A common approach to anticipate SFOs is to focus on a specific predictability source, e.g., strong tropical heating associated
with ENSO or the MJO. However, many potentially predictable signals may be evolving at any given time, and the constructive
or destructive interference between each signal's teleconnections may enhance or degrade the overall predictable forecast
signal for variables that we are interested in, e.g., southwest Asian precipitation. Thus, it is more desirable to use a method that
considers *all* relevant signals, and their combined influence, to anticipate the overall likelihood of a skillful forecast over the
region of interest. Here, following Sardeshmukh et al. (2000), the theoretical expected skill of a perfect, infinite ensemble
member forecast, $\rho_\infty(\tau, t)$ (Eq. (5)), is selected to identify SFOs, based on the method's past success. In particular, expected
skill is calculated using the pattern correlation version of the LIM signal-to-noise ratio, $S^2$ (Eq. (6); Newman et al. 2003)
evaluated over the southwest Asian domain is calculated at each forecast lead time $\tau$, and each initialization date t. As a result,
$S^2$ and expected skill are a function of time, but not space:

$$\rho_\infty(\tau, t) = \frac{S^2(\tau)}{\{[S^2(\tau)+1]S^2(\tau)\}^{.5}} \quad \{5\}$$

$$S^2(\tau, t) = \frac{tr[F(\tau)]}{tr[E(\tau)]} \quad \{6\} \qquad .$$





$S^2$ is determined using $\mathbf{F}(\tau, t)$, the forecast signal covariance matrix determined at a given lead time, which indicates the strength of the predictable signal in the forecasts, and $\mathbf{E}(\tau)$, the forecast error covariance matrix which represents lead-dependent, unpredictable 'noise':


$$\mathbf{F}(\tau, t) = <\hat{\mathbf{x}}(t+\tau)\hat{\mathbf{x}}(t+\tau)'> \qquad \{7\}$$

$$\mathbf{E}(\tau) = \mathbb{C}0 - \mathbf{G}(\tau)\mathbb{C}0\mathbf{G}(\tau)' \qquad \{8\}.$$

Where $'$ denotes the matrix transpose. Note that $\mathbf{E}$ is not a function of time, consistent with the assumption of state-independent noise (Eq. (2)), but does vary with forecast lead time $\tau$.

We define SFOs as the top 20% of expected skill forecasts, as this subset accurately identifies significantly more skillful forecasts for a range of lead times (Fig. 1). ACC and PCC during these dates are compared to the skill when, instead of expected

skill, the top 20% of Niño3.4 amplitude and RMM amplitude are used to identify periods of elevated skill. The skill during these three subsets of forecasts (expected skill, Niño3.4, RMM) is compared to the skill of the remaining 80% of expected skill forecasts, and the 95% confidence level in the skill differences during the three subsets is assessed nonparametrically, using bootstrapping with replacement. Similar skill differences were found for a range of 10-25% of the forecasts in the SFO group, and 20% was chosen because it provided the greatest number of samples – useful for further separating forecasts by ENSO

and MJO phase later – but small enough so that the subset has significantly elevated skill.

### 2.2.2 Relative Risk

A relative risk ratio is used to quantify shifts in the likelihood of an SFO occurring as a function of ENSO, MJO, and SPCZ OLR strength. This is done, for each index, by determining the fraction of SFOs initialized on days with index values of varying amplitude:


$$\text{FRAC} = \# \text{ SFOs} / \# \text{ dates} \qquad \{9\}$$

For ENSO and SPCZ OLR, changes in SFOs during positive and negative index values of varying threshold are considered, while the RMM is always positive. Instead, we assess changes in SFOs during MJO phases 2-3 or 6-7 and increasing RMM

thresholds. To determine the relative risk of an SFO compared to the probability of one occurring on any random day, we divide FRAC calculated for each threshold and index group by 0.2 – since for top 20% of expected skill dates, the chance of one occurring on any date in the forecast period is 0.2:



$$\text{Relative Risk} = \text{FRAC} / 0.2 \qquad \{10\}$$


As an example, for weeks 3-4 SFOs when Niño3.4 > 1.5, 117 SFOs are found during the 269 days exceeding that threshold, corresponding to FRAC = 0.44 and Relative Risk = 2.2. The robustness of these estimates is evaluated by determining the 95% confidence bounds around the relative risk estimates using bootstrapping with replacement.

## 3 Results

Section 3.1 shows how expected skill can stratify skillful and unskillful forecasts to identify SFOs and demonstrates how strong ENSO and MJO phases increase the likelihood of an SFO occurring. Section 3.2 reveals how anomalous SPCZ OLR is observed during both wet and dry SFOs, and how predictable patterns during El Niño and La Niña conditions are associated with unique circulation features.

### 3.1 Identifying Forecasts of Opportunity

The distribution of LIM forecast skill, measured by PCC during the top and bottom 20% of theoretical expected skill forecasts (Eq. (5)), confirms that the high expected skill group successfully identifies more skillful forecasts than the low expected skill group (Fig. 1). Note that for each lead time, the high and low expected skill dates identified are not necessarily the same. For lead times of weeks 2-4, both the median and 95[th] percentile values of the PDFs of forecasts initialized on high expected skill dates show statistically significant shifts towards higher PCC, with the greatest skill increases, relative to the bottom 20% 210 group, at the shortest lead time of two weeks (Fig. 1a-c). The distribution of week 2 PCC is also the narrowest for the high expected skill group, a reflection of the more deterministic nature of forecasts at this lead time, particularly during periods of high signal-to-noise ratio (Eq. (6)). As lead time increases, the distribution of skill widens as forecast uncertainty increases, so that by week 5, the medians are indistinguishable between the two PCC distributions. Still, some skillful forecasts remain at week 5 in the high expected skill group, shown by the statistically significant shift in the 95[th] percentile of PCC.


Subseasonal precipitation skill, evaluated using ACC for weeks 3-4 and 5-6, is low, as discussed in past studies, but increases substantially during high expected skill periods (Figs. 2-3). The LIM 'All Dates' weeks 3-4 skill of .2-.3 ACC exceeds the week 3 skill of most of the S2S models evaluated by de Andrade et al. (2018) for November – March 1999-2009 (compare Figs. 2a, to their Fig. 1). Comparing the three approaches to anticipating SFOs – high expected skill, Niño3.4, and RMM – 220 expected skill most successfully anticipates SFOs at both weeks 3-4 and 5-6. The location of maximum skill shifts southeastward from weeks 3-4 to 5-6, with skill also weakening at longer lead times as expected. While there are some regions experiencing a skill increase during the top 20% of Niño3.4 and RMM amplitude dates, the increases are mainly indistinguishable from the skill of the remaining forecasts (Fig. 2c,d, 3c,d). Splitting the Niño3.4 index to consider only strong El Niño or La Niña events indicates that some regions do experience elevated skill during both phases, though in different,



localized regions that only cover a limited portion of the region compared to the forecasts identified using expected skill (Figure S1).

Considering PCC during the high expected skill, Niño3.4 and RMM dates confirms that forecasts initialized during periods of high expected skill generally have higher PCC than those identified using Niño3.4 and RMM (Fig. 4). For lead times between

weeks 2-4, median PCC shifts statistically significant shifts at the 95% confidence level reflect the increase in skill, as do the increased probability density of forecasts with PCC > 0.5. By week five, the distributions of PCC during high expected skill and high Niño3.4 dates become indistinguishable, consistent with greater similarity in the regions of skill found using Niño3.4 and expected skill at weeks 5-6 (Fig. 3). Overall, the expected skill metric is more effective at anticipating SFOs than Nino3.4 and RMM.

**3.1.1 Relating SFOs to Niño3.4 and RMM Indices**

The advantage of using high expected skill to anticipate SFOs is that it measures when the combined signal of *all* forcings (e.g., ENSO, MJO) is high relative to unpredictable 'noise'. High expected skill occurs during periods of constructive interference between signals, while low expected skill reflects periods of deconstructive interference (e.g., Farrell 1988; Farrell and Ioannou 1996; Albers and Newman 2019), and such interference is more intermittent than the individual forcing elements

(Figs. 5, 7). As such, despite what is shown in Figures 2-4, many high expected skill dates occur during strong ENSO and MJO events, as indicated by the overlay of SFOs (black dots) with time series of Niño3.4 (Fig. 5) and RMM (Fig. 7). The bottom 20% of expected skill forecasts are also shown in the vertical light gray lines, to contrast the higher-frequency expected skill with the lower-frequency Niño3.4 and RMM. The correspondence to Niño3.4 is considered first. Both El Niño events of 1983 and 2016 coincided with high expected skill dates at weeks 3-4 and 5-6, though on different dates; conversely, the 1998

event was not associated with any high expected skill dates at weeks 5-6 but was for weeks 3-4. Strong La Niña events, such as 1999, also reflect periods of high expected skill. Still, there are many high expected skill forecasts initialized with weak Niño3.4 values, as in 2017, since other processes – including ENSO-related heating not captured by Niño3.4 – can produce a high signal-to-noise ratio. Moreover, some of the lowest expected skill dates occur during strong ENSO events, such as in 2016 for weeks 3-4 at the beginning of February, suggesting other processes may have been destructively interfering with the

ENSO-related component.

High Niño3.4 index amplitude during both El Niño and La Nina events leads to increases in the risk of SFO occurrence, though more strongly during the former than the latter (Fig. 6). There is a greater relative risk for SFOs in weeks 5-6 than weeks 3-4, suggesting that at longer lead times within the subseasonal forecast period, ENSO conditions are increasingly important for

SFOs. However, it is important to note that skill during these periods is overall lower than weeks 3-4 (Figs. 2-3), which could be due to the limited capability of ENSO alone to impact predictability, and/or elevated noise. The asymmetric response during El Niño and La Niña conditions may reflect the fact that there are more frequent high amplitude El Niño events than La Niña





events, increasing the number of samples at higher Niño3.4 thresholds. Indeed, Niño3.4 exceeds 1.5°C on 269 days, compared to 192 days where Niño3.4 is less than -1.5°C. The stronger response during El Niño conditions than La Niña could also be
consistent with Hoell et al. (2018a), who found precipitation shifts during both CP and EP El Niño events but only CP La Niña events, although future work is needed to better explore these nuances.

Discerning a relationship between MJO phases 2-3 and 6-7, the phases that impart known teleconnections to southwest Asian precipitation (Hoell et al. 2018b), can be more difficult given the more transient nature of the MJO compounded with transient
expected skill (Fig. 7). Only the relationship between the RMM and weeks 3-4 expected skill is considered, as an MJO teleconnection at 5-6 weeks lead times is not physically plausible (e.g., Tseng et al. 2018). Still, we do find that particularly strong events for phases 2-3 and 6-7 increase the relative risk of weeks 3-4 SFOs, though sampling introduces spread in these estimates (Fig. 8). Some particularly high-amplitude MJO events, including phases 2-3 in 1985 and 1994 and phases 6-7 in 2005 and 2018, overlap with periods of weeks 3-4 high expected skill, while some weaker amplitude events overlap with some
of the lowest expected skill dates, such as phases 2-3 in late January 2002. Strong RMM phases 6-7, also increase the relative risk of SFOs, which is elevated at lower RMM thresholds compared to phase 2-3 and does not display the exponential increase at the highest thresholds. These subtle differences in relative risk sensitivity could reflect true differences in the MJO teleconnection to the region, or could be due to sampling, as there is high uncertainty in the relative risk estimates given the small number of events observed at such high thresholds.

### 275 3.2 Characteristics of Predictable Dry and Wet Initializations

This section compares the composite patterns preceding predictable wet and dry periods 18 days earlier, revealing the role for anomalous heating near the SPCZ. Next, composite wet and dry periods are split by Niño3.4 sign, revealing how different circulation responses during each phase produce like-signed precipitation anomalies over southwest Asia.

#### 3.2.1 All Initializations

First, we consider the patterns preceding anomalously wet and dry periods regardless of ENSO phase, where 'wet' and 'dry' are defined using the top and bottom terciles of southwest Asian precipitation anomalies, respectively. Wet and dry dates that were associated with weeks 3-4 high expected skill dates initialized 18 days earlier, and were also characterized by a PCC > 0, are considered. A lead time of 18 days is chosen because it falls within the weeks 3-4 forecast period and provides the clearest circulation structures, which are similar but weaker at longer lead times (not shown). For these skillful, high expected
skill forecasts associated with the development of anomalously wet or dry precipitation anomalies, we consider the composite circulation and heating patterns observed at the time of initialization, or 18 days before the anomalous precipitation is observed.

Figure 9 shows that during weeks 3-4 SFOs initialized before predictable wet and dry periods over southwest Asia, anomalies are roughly equal and opposite in sign. Before dry periods, positive OLR anomalies are located over the western and central





tropical Pacific, signifying suppressed convection, while a 200-hPa anticyclone (positive $\Psi_{200}$ anomaly) is located over

southwest Asia, consistent with downward vertical motion and suppressed precipitation. Conversely, predictable anomalously

wet periods are associated with enhanced anomalous central Pacific convection and a cyclonic $\Psi_{200}$ anomaly over southwest

Asia. One feature present during dry initializations is a small but statistically significant negative OLR anomaly in the eastern

Indian ocean, which does not have a counterpart during wet initializations. Southwest Asian precipitation is strongly linked to

heating variability in this location (Hoell et al. 2012), and the results here suggest that the relationship might be particularly

important for anomalously dry periods.

### 3.2.2 El Niño and La Niña Initializations

Section 3.1 indicated that, while anticipating SFOs based on the Niño3.4 index alone is less successful than expected skill

(Figs. 2-3), periods of strong ENSO activity increase the likelihood a forecast of opportunity occurs, particularly – intuitively

– during the strongest events (Fig. 6). While during either ENSO phase, periods of anomalously high or low precipitation can

occur due to transient, synoptic-scale disturbances forming along the subtropical jet, how the precipitation anomalies develop

differs between El Niño and La Niña, given ENSO's influence on the mean jet and circulation (Shapiro et al. 2001; Henderson

et al. 2020). This section examines how predictable wet and dry initializations differ by ENSO phase, given the hypothesized

differences in teleconnections in each phase.


Splitting dry and wet forecast initializations into periods when Niño3.4 is positive or negative to reflect El Nino or La Niña

conditions, without losing any samples, indicates that even when preceding same-signed precipitation anomalies, El Niño and

La Niña conditions are associated with different large-scale circulation patterns (c.f., Fig. 10a,c; Fig.10b,d). By construction,

there are clear differences in SST and OLR associated with El Niño and La Niña conditions, as well as north Pacific $\Psi_{200}$

anomalies associated with ENSO-like tropical heating (Winkler et al. 2001; Henderson et al. 2020). During La Niña conditions,

dry periods include an anticyclonic anomaly over southwest Asia, while during dry El Niño initializations there cyclonic

features north and east of southwest Asia and weak anomalies directly over the region (c.f. Fig. 10a,c). In contrast to the

circulation pattern during dry El Niño periods (Fig. 10c), during wet periods, El Niño conditions are associated with an

amplified $\Psi_{200}$ pattern, with two high amplitude cyclonic anomalies located over Eurasia (Fig. 10d). Conversely, at the time

of initialization during wet La Niña periods, negligible $\Psi_{200}$ anomalies are observed (Fig. 10b).

What distinguishes rainy La Niña or El Niño periods from dry La Niña or El Niño periods? While the heating and SST dipole

patterns are consistent for both wet and dry periods of each ENSO phase, there are differences in heating strength and location

(c.f. Fig. 10a,b; Fig.10c,d). Dry La Niña initializations include stronger negative SST anoamlies and suppressed convection in

the central Pacific compared to rainy periods, which instead involve enhanced convection over the maritime continent. Dry El

Niño periods include stronger suppressed convection over the maritime continent than wet El Niño dates, coinciding with a

hint of a wave train emanating from the east Pacific across North America and the north Atlantic. Dry La Niña dates are more





common than wet La Niña dates, 84 vs 67 days, while wet El Niño dates are more common than dry El Niño dates, 96 vs 57 days, consistent with past research linking seasonal mean departures of southwest Asian precipitation to ENSO (Hoell et al.

2018a).

Differencing the dry and wet composites between each ENSO phase reveals the common element of suppressed SPCZ convection and cooler central Pacific SSTs during dry periods (Figure 11). Dry periods during El Niño conditions are associated with warmer SSTs in the east Pacific than wet periods (Fig. 11b), while no such differences in SSTs or OLR are

observed in the east Pacific during La Niña conditions. While distinct from one another, the $\Psi_{200}$ patterns during both El Niño and La Nina conditions both place an anomalous anticyclone over southwest Asia during dry events, consistent with suppressed precipitation. The $\Psi_{200}$ patterns during dry versus wet periods differ, with La Niña conditions displaying weak anticyclonic anomalies in the subtropical north Pacific and north Atlantic, and El Niño conditions associated with an upper-level wave train emanating from the eastern tropical Pacific, across the north Atlantic to Europe, potentially linked to the anomaly over

southwest Asia. The orientation of such a wave train is consistent with the evolution described by Shaman and Tziperman 2005, who found a northeastward propagating wave train emanating from the east-central Pacific during strong ENSO events, which ultimately modulated Tibetan snow depth. Thus, while the heating difference between dry and rainy periods is similar regardless of ENSO phase, the impact of the anomalous heating on the circulation is different, but coincidentally yields a reduction in precipitation over southwest Asia. The different circulation responses are consistent with the modified mean states

of each ENSO phase, though more work is required to further understand these nuanced relationships, preferably with a larger sample size.

### 3.2.3 Relative Risk associated with SPCZ OLR

Given the similar OLR anomalies in the SPCZ region noted during predictable wet and dry forecast initializations (Fig. 9; Fig. 11), a time series of OLR over the region was selected for a final metric to consider related to week 3-4 SFOs. Similar to

considering Niño3.4 and RMM, using OLR alone to anticipate periods of elevated skill, the relative risk of a high expected skill date increases significantly as the standard deviation of SPCZ OLR anomalies increases (Fig. 12). The response during negative and positive OLR anomaly values is more symmetric than the risk associated with increasing Niño3.4 threshold, which indicated a greater relative risk increase during El Niño than La Niña conditions (Fig. 6) or comparing the impact of MJO phases 2-3 vs 6-7 (Fig. 8). The SPCZ OLR time series is correlated with the Niño3.4 index at r = -0.25, an indication that

the SFOs associated with SPCZ OLR are not redundant with Niño3.4-related SFOs and are therefore contain additional information about SFOs related to tropical variability. As such, the expected skill approach to SFOs benefits from measuring shifts in the likelihood of a forecast of opportunity captured by several distinct indices tracking tropical variability, Niño3.4, RMM and SPCZ OLR, a distinct advantage over using an index that tracks only of these processes (Figs. 2-3).



## 4 Conclusions

In this study, precipitation SFOs are considered over southwest Asia using LIM expected skill, a metric related to the forecast
signal-to-noise ratio that leverages the constructive interference of all signals impacting predictability. Strong El Niño, La Nina
and MJO phase 2-3 and 6-7 conditions increase the chances that an SFO occurs. A third tropical heating index, based on
anomalous OLR in the SPCZ region, also increases the risk of an SFO (Fig. 12). The correspondence between expected skill
and several indices highlights the advantage of using expected skill, in that all of these flavors of tropical heating are registered

as high signals. However, there are still SFOs that do not correspond to any one of these indices, since other processes,
potentially not tropically-driven, can produce a high signal too. Future work could focus on categorizing all SFOs to examine
these potential additional factors.

In addition to the confirmed influence of ENSO and MJO activity on southwest Asian precipitation, anomalous heating along

the SPCZ region is a common element among predictable wet and dry initializations and increases the relative risk of an SFO.
Heating in this region is also associated with different circulation patterns during El Niño and La Niña conditions (Fig. 11).
How these different circulation patterns are related to similar anomalous heating anomalies is currently not well understood,
but likely involves the modified tropopause-level waveguide present during each ENSO phase, which modulates the
extratropical response to tropical heating (Sardeshmukh and Hoskins 1988; Newman and Sardeshmukh 1998; Shapiro et al.

2001). It's also important to note that, while widely used, ENSO indices such as Niño3 or Niño3.4 do not capture the full
spectrum of ENSO variability (Penland and Sardeshmukh 1995; Newman et al. 2009; Gehne et al. 2014; Henderson et al.
2020). Future work could employ the dynamical decoupling approach of Henderson et al. (2020) to isolate the ENSO signal
and its impact on precipitation SFOs more holistically.

The association between forecasts of opportunity and the MJO is less constrained given the higher-frequency nature of the
MJO and small sample size once RMM is sorted by phase, but still indicates a role for strong MJO events in phases 2-3 or 6-
7 to increase the likelihood of a weeks 3-4 SFO occurring, consistent with prior studies (Cannon et al. 2017; Hoell et al. 2018b).
Further suggesting a role for MJO-like heating, predictable heating patterns associated with forecasts of opportunity indicate
a role for anomalous convection over the Indian ocean during dry periods, consistent MJO phases 2-3 suppressing southwest

Asian precipitation (Fig. 9a). Future work could employ large climate simulation output to enhance sample size and revisit the
MJO/expected skill relationship, to the extent the model can reproduce the mean state and MJO itself. Another remaining
question that could be addressed more aptly with a larger sample size is how ENSO and the MJO act together to impact SFOs
for southwest Asian precipitation events, particularly concerning their magnitude and duration, which was beyond the scope
of this study but merits further investigation.


## Data Availability



The JRA-55 Reanalysis data used in this study is freely available at https://rda.ucar.edu/datasets/ds628.0/, and CHIRPS precipitation is freely available at https://data.chc.ucsb.edu/products/CHIRPS-2.0/global_daily/netcdf/p25/. The RMM index was accessed at no cost from the Australian Bureau of Meteorology here:

http://www.bom.gov.au/climate/mjo/graphics/rmm.74toRealtime.txt.

**Author Contributions**

Melissa L. Breeden wrote code for calculations, produced all figures and wrote the manuscript. John R. Albers provided technical expertise on the LIM and subseasonal forecasting, frequent guidance on computations and figures, and provided edits

to the manuscript. Andrew Hoell secured funding for this project, provided expertise on southwest Asia, frequent guidance on figures, and edits to the manuscript.

**Competing Interests**

The authors declare no conflict of interest.


**Acknowledgements**

The authors gratefully acknowledge support from the Famine Early Warning Systems Network.

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

| Variable | Domain | % Variance Explained | # EOFs Retained |
|---|---|---|---|
| SST | 20°S-20°N, 0-357.5°E | 63% | 8 |
| OLR | 20°S-20°N, 0-357.5°E | 54% | 23 |
| $\Psi_{200}$ | 0-90°N, 0-357.5°E | 63% | 10 |
| Precip | 15-48°N, 40-80°E | 70% | 10 |
| 2mT | 15-45°N, 40-90°E | 77% | 5 |

Table 1. LIM variables that compose the state vector **x**, their domain, and the % variance explained by the retained EOFs. Variables include 200-hPa streamfunction ($\Psi_{200}$), 2-meter temperature (2mT), sea surface temperature (SST), and outgoing longwave radiation (OLR). Precipitation (Precip) from the Climate Hazards InfraRed Precipitation with Stations (Funk et al.
2015) Version 2.0 dataset (https://data.chc.ucsb.edu/products/CHIRPS-2.0/) is also used. JRA-55 and CHIRPS are used because they are available in near-realtime (2-3 day lag) and serve as the basis for experimental realtime forecasts, and are available back to at least 1981. $\Psi_{200}$, SST and OLR anomalies are used on a 2.5 X 2.5 degree horizontal grid, while 2mT and Precip are used on a 0.5 X 0.5 degree grid. The EOFs and PCs retained in **x** are not sensitive to the gridding used (not shown).





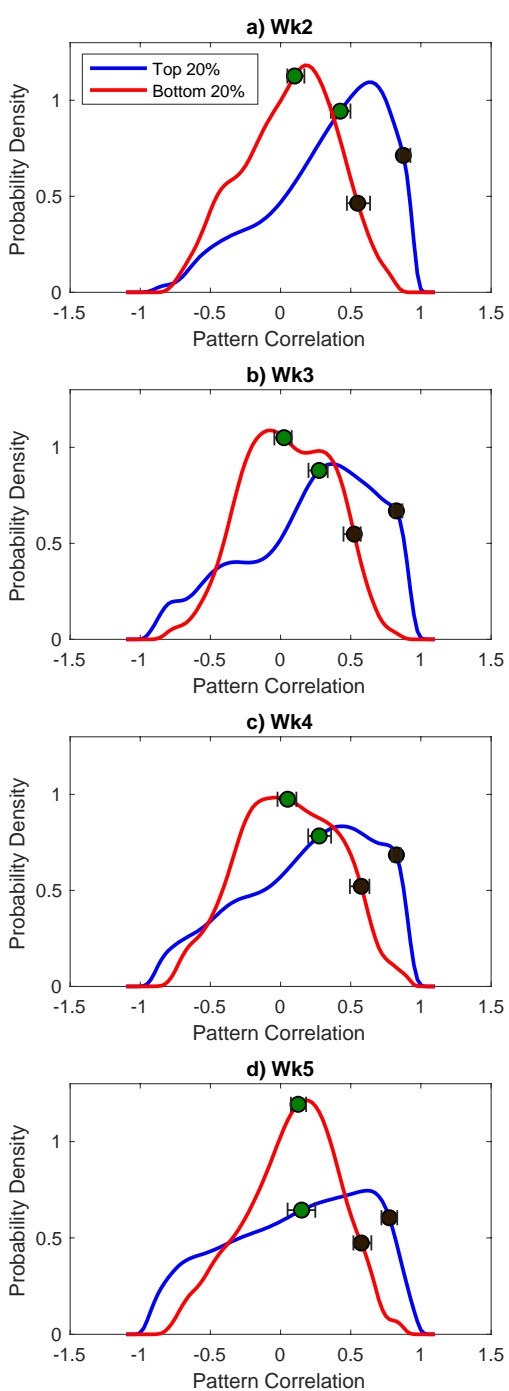

**Figure 1: PDFs of actual skill measured by pattern correlation (PCC), for the top 20% and bottom 20% of expected skill dates for weeks of lead times a) two – d) five. The green circles represent the bootstrapped median values, and the black circles represent that bootstrapped 95th percentile values.**




## Weeks 3-4 ACC, Precipitation

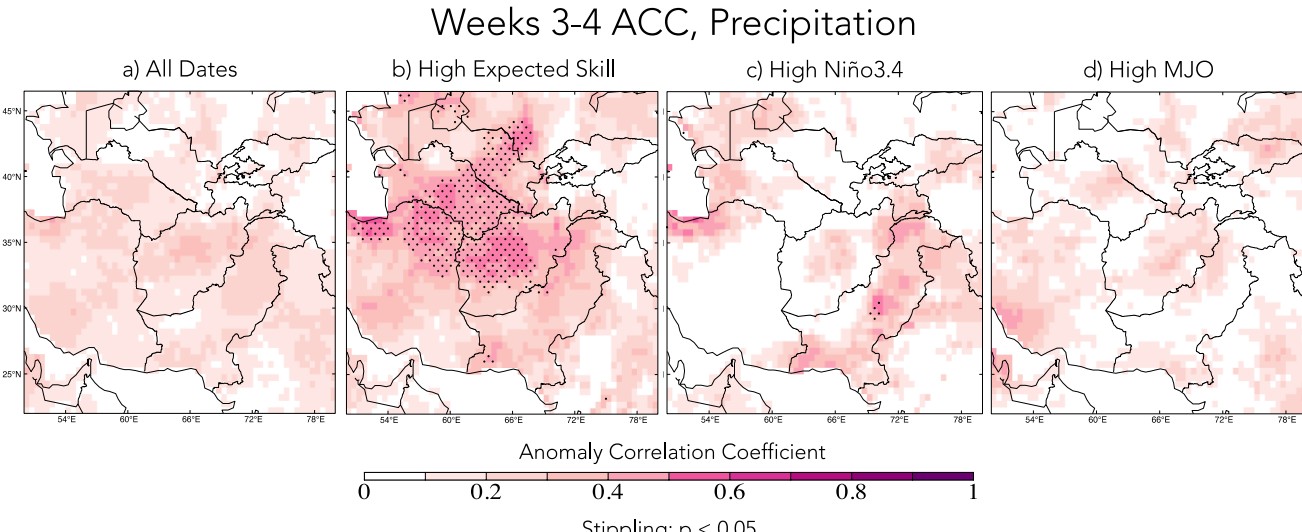

**Figure 2: Anomaly correlation coefficient (ACC) for weeks 3-4 forecasts, evaluated from January – March 1982-2020. Panel a) shows ACC for all dates in the record, b) ACC for the 20% of forecasts initialized with the highest expected skill, c) ACC for the 20% with the highest Niño3.4 amplitude, and d) ACC for the 20% with the highest RMM amplitude. The black stippling indicates where the skill of the top 20% of forecasts in each group is statistically significantly different from the skill of the remaining 80% of forecasts at the 95% confidence level, determined nonparametrically with bootstrapping.**

## Weeks 5-6 ACC, Precipitation

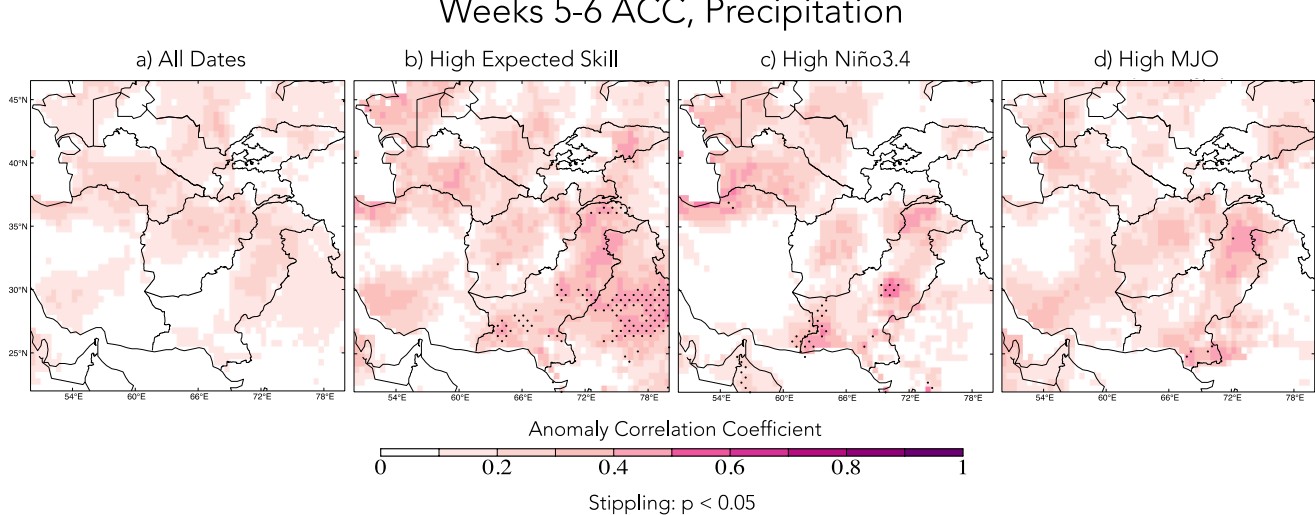

**Figure 3: As in Figure 2 but for weeks 5-6 forecasts.**




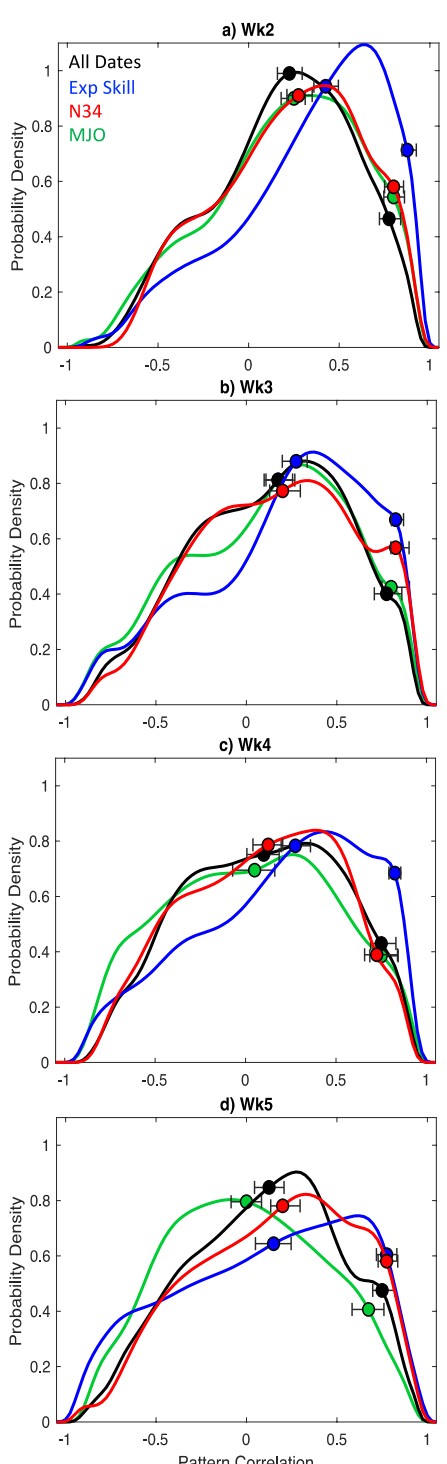

**Figure 4: PDFs of precipitation PCC for all forecasts (black lines), the 20% of forecasts with highest expected skill (blue), Niño3.4**
595 **(red), and RMM (green) initializations. The bootstrapped confidence intervals for the median and 95th percentile of the distribution are shown with the markers.**





**Figure 5: The color shading shows the Niño3.4 index and is the same in both panels. The black dots indicate the top 20% of expected skill forecasts, and the gray vertical lines represent the bottom 20% of expected skill forecasts, for weeks 3-4 forecasts (left) and 600 weeks 5-6 forecasts (right).**




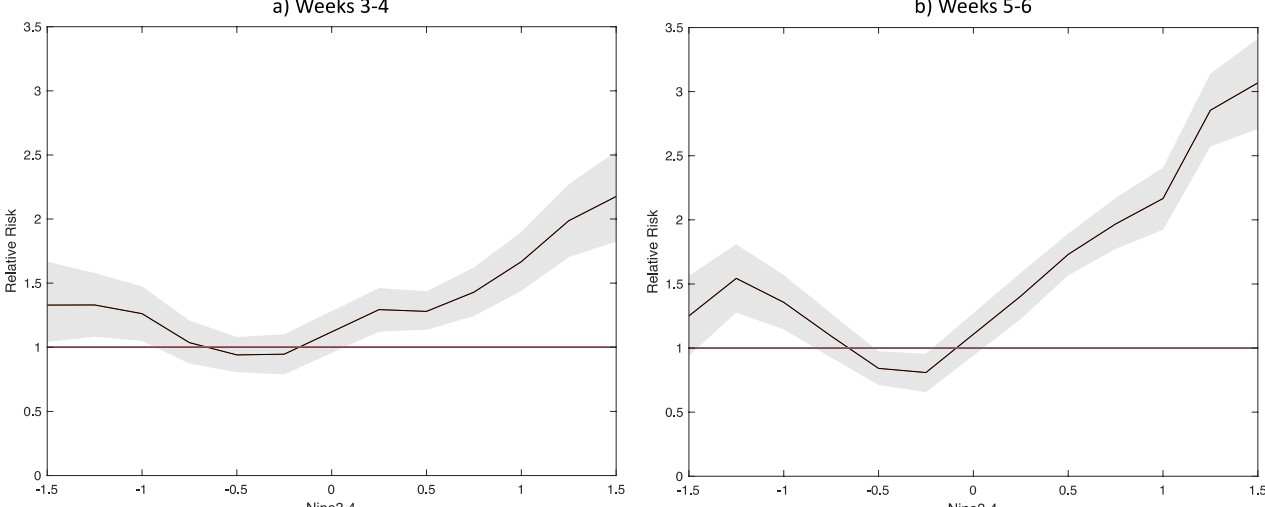

**Figure 6: The black line shows the relative risk, relative to the risk on any given day, of an SFO, meaning one of the top 20% of expected skill dates, occurring when the Niño3.4 index is greater than various thresholds, with 95% confidence intervals in gray shading.**



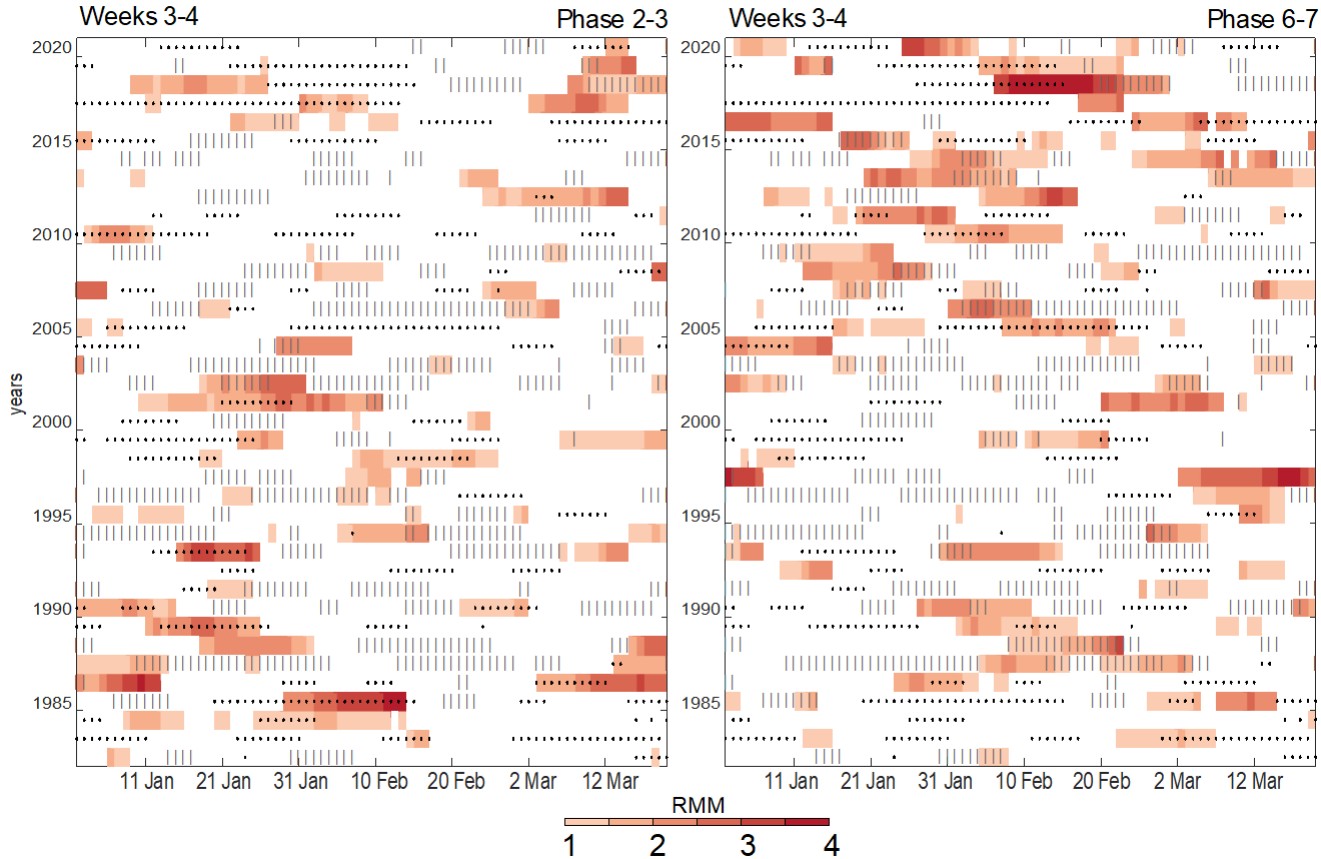

**Figure 7: As in Figure 5 but for the amplitude of the RMM index during a) MJO phases 2-3 and b) MJO phases 6-7 and for weeks 3-4 expected skill dates.**





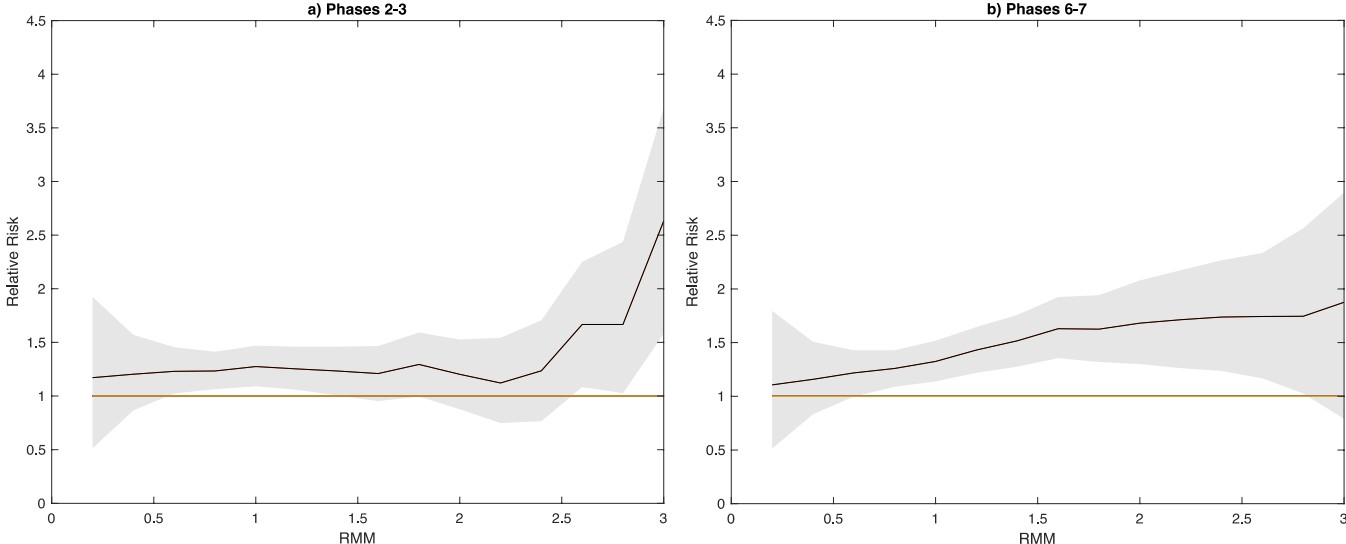

**615 Figure 8: As in Figure 6 but for weeks 3-4 expected skill and for varying RMM amplitude during either MJO phases a) 2-3 or b) 6-7.**

620



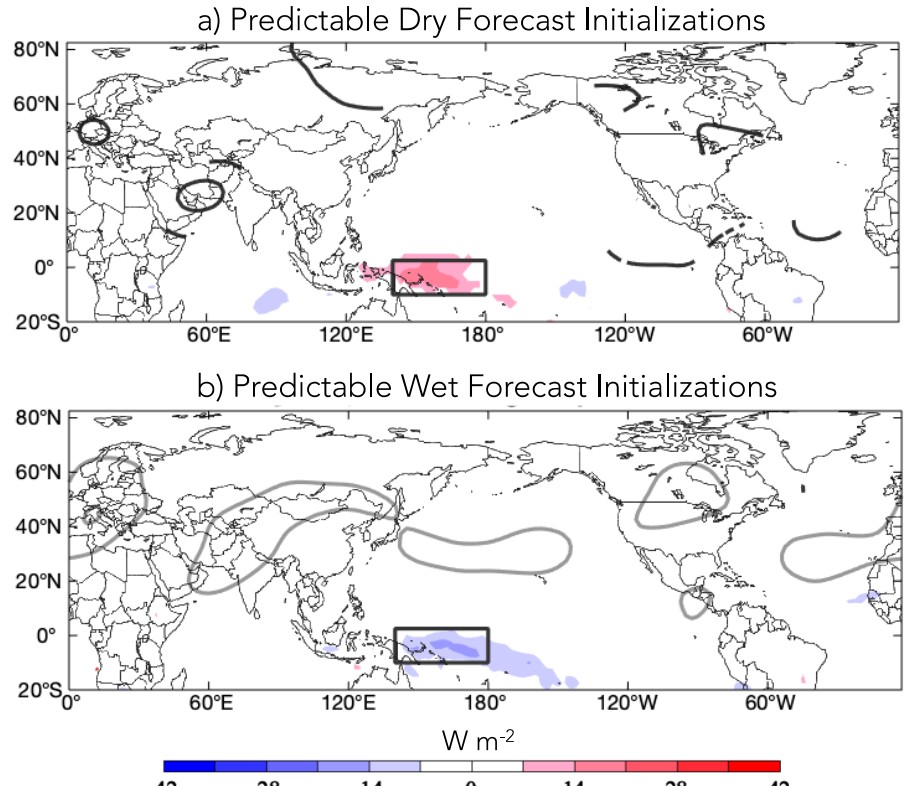

**Figure 9: Composite OLR (color shading), SST (red/blue ontours), and $\Psi_{200}$ (black/gray contours) anomalies, during a) high expected skill initializations verifying on anomalously dry days 18 days later, N = 142 days. Panel b) shows high expected skill initializations verifying on anomalously wet days 18 days later, N = 163. Negative SST anomalies are shown in blue contours and positive SST anomalies are shown in red, at a contour interval of 0.5 degrees C. $\Psi_{200}$ anomalies are plotted positive (solid black) and negative (solid gray) at a contour interval of $4*10^6$ m$^2$ s$^{-1}$. Full-field anomalies are shown, while the LIM forecasts are initialized on the EOF-truncated representation of the anomalies in the state vector (Table 1). Only anomalies that are statistically significant at the 95% confidence level are shown. The significance of anomalies was determined nonparametrically using bootstrapping with replacement.**





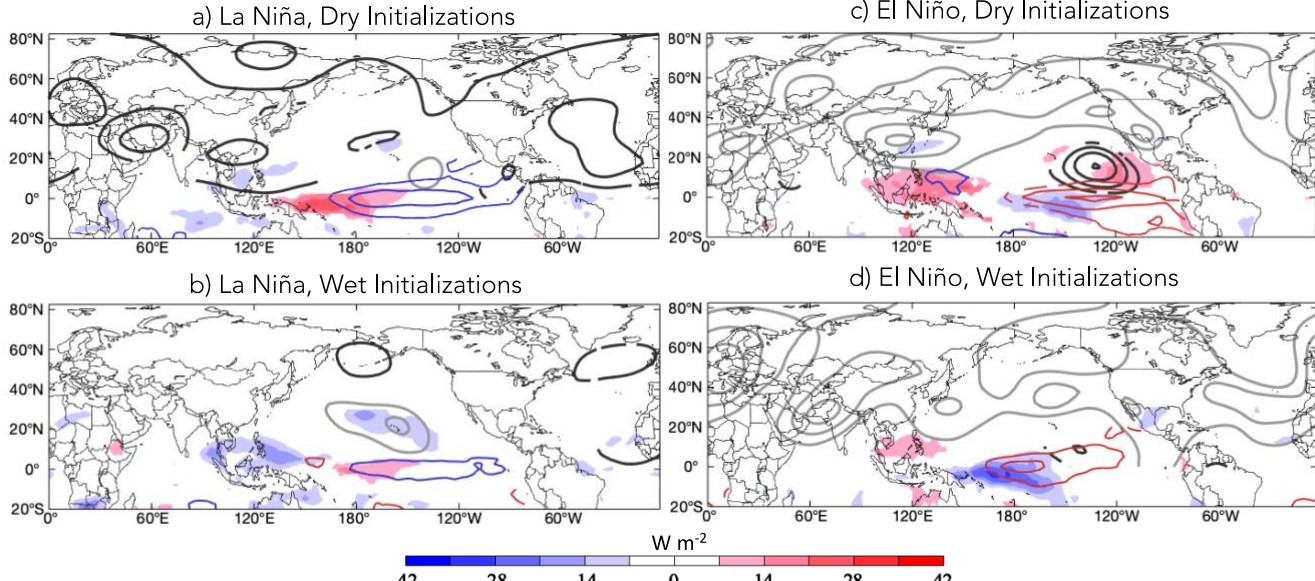

**Figure 10: As in Figure 9 but for groups divided by Niño3.4 < 0 ('La Niña') or Niño3.4 > 0 ('El Niño').**



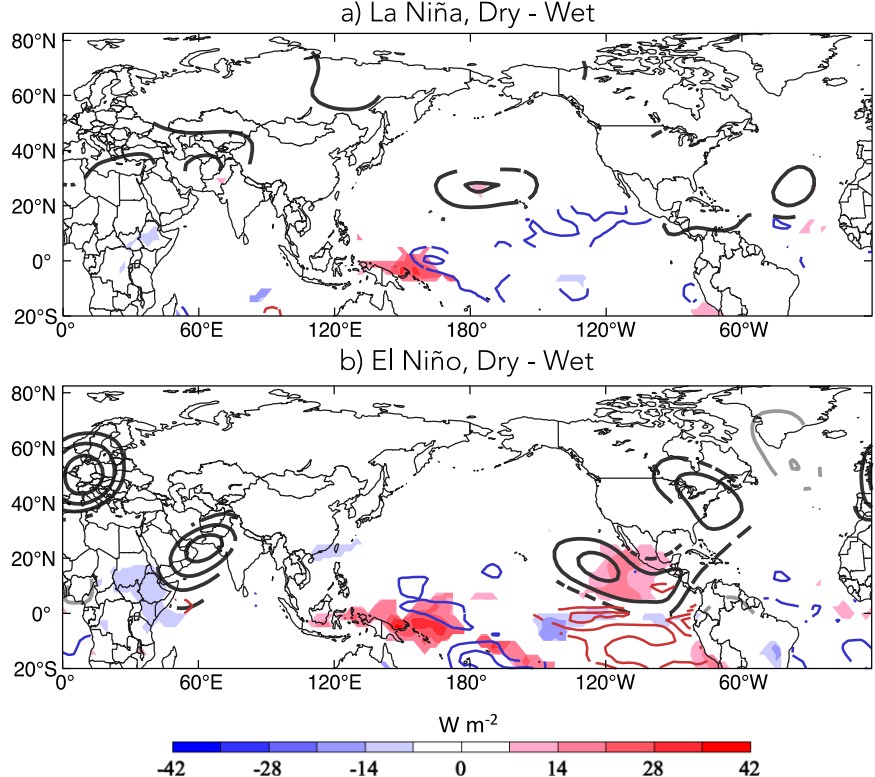

**Figure 11:** Composite difference, dry – wet, during a) La Nina conditions (Fig. 10a-b) and b) El Niño conditions (Fig. 10c-d). Plotting conventions are as in Figure 9, except the contour interval for SST anomalies (blue/red contours) is 0.25 degrees Celsius.

645





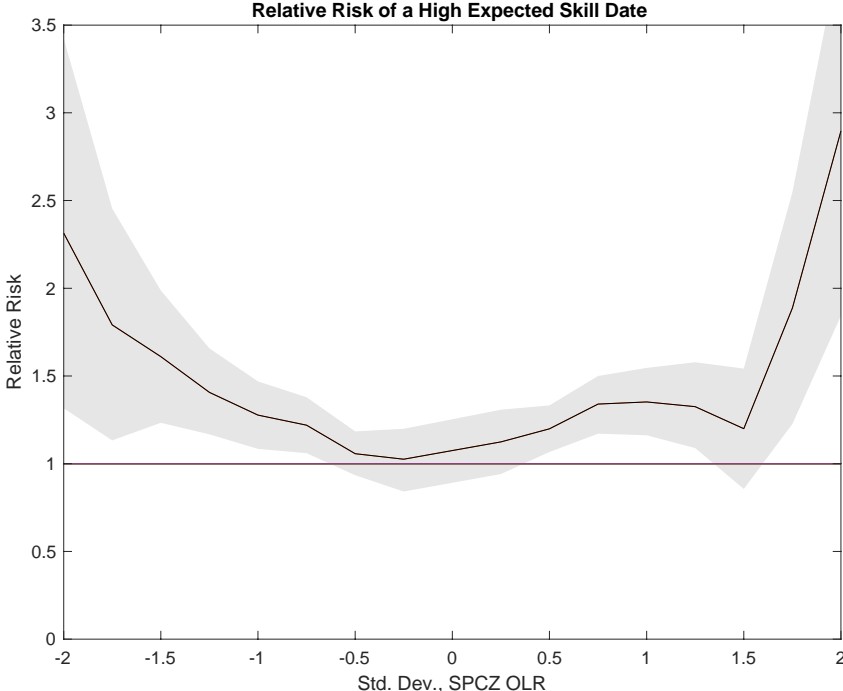

**Figure 12: As in Figure 6 but for using the standard deviation of the SPCZ OLR anomaly time series, calculated using the boxed region in Figure 9 and for weeks 3-4 expected skill.**

650