# Peer review of "Subseasonal precipitation forecasts of opportunity over central southwest Asia"

_EGUsphere, 2022_

## Referee Comment (RC1)

**Review of manuscript WCD-2022-555 entitled "Subseasonal precipitation forecasts of opportunity over southwest Asia" by Melissa Leah Breeden, John Robert Albers and Andrew Hoell**

**OVERALL RECOMMENDATION**

Minor revision

**SUMMARY**

This study uses a statistical model to forecast precipitation at the subseasonal timescales (weeks 3-4 and 5-6) over a specific target region (southwest Asia). The statistical model is a Linear Inverse Model (LIM), and the main goal of the manuscript is to illustrate how some intrinsic characteristics of those LIM forecasts (based on signal-to-noise ratio) enable to "forecast the forecast skill" (Kalnay and Dalcher, 1987). The authors show that the LIM expected skill is a powerful tool to detect forecasts that are indeed truly skillful. Additional analysis shows that this expected skill indicator is more relevant to detect forecasts of opportunity than indices related to ENSO and the MJO. Finally, it also enables to identify another physical signal, namely OLR anomalies in the SPCZ, that is more closely related to these forecasts of opportunity.

**MAJOR COMMENTS**

This study builds on the LIM methodology, that has already been used by some co-authors of the current manuscript to identify subseasonal forecasts of opportunity (e.g Albers and Newman 2019). To my mind, the major advances that it proposes relative to previous LIM studies are:
i) the forecast of a local meteorological parameter, i.e precipitation. This makes this study very relevant on the road to real-time and user-oriented applications of subseasonal forecasts
ii) the comparison with usual climate indices (ENSO and MJO)
iii) the identification of the SPCZ OLR anomalies as an additional physical indicator of forecasts of opportunity
Moreover, the manuscript is clearly written and provides interesting discussions on the "forecast of opportunity" concept. I think it will be fit for publication once the authors have addressed the minor revisions and my main concern below.

My main concern is about the description of the LIM forecast setup, which I feel is rather incomplete:
1) What is the frequency of initialization during the 1 January – 20 March period? Do you start a LIM forecast everyday? Every week?
2) I do not completely understand what the output of your LIM forecast is exactly. My assumption would be that it provides weekly values because it is fitted on weekly averages, but I am not certain. Then, where do the biweekly forecasts come from?

For instance, do you add the weekly forecast at day 15 and 22 in order to get week 3-4 precipitation?

3) Why do you mix up verification on weekly (Figures 1 and 4) and biweekly values? I would have expected all results to be shown on biweekly values.

4) By the way, you should be more explicit about the convention you use for week 2, week 3, etc.

All in all, I think some figure illustrating the outline of the LIM forecasting process for a specific example would be welcome.

**MINOR COMMENTS**

l. 23: Could you include a few words to help the reader visualize what you mean by "southwest Asia" (e.g names of countries or geographical features such as the Persian Gulf, the Caspian Sea, the Himalaya)?

Figure 1: Could you also show the distribution of PCC when considering all LIM forecasts (and not only the top or bottom 20% of expected skill)?

You could explicitly specify in the introduction that no numerical subseasonal forecast (e.g S2S, SubX) is used in this study, contrary to other previous studies such as Albers and Newman (2019).

**TYPOS**

l.299-300: "particularly – intuitively – during the strongest events"
There seems to be one adverb too many, at least for the clarity of the sentence. I suggest rephrasing.

l. 300-302: "While during either ENSO phases, (…) on the mean jet and circulation".
I do not understand this sentence, which seems too long. Isn't there a missing verb? I suggest rephrasing.

l. 306 and elsewhere: "El Nino" → "El Niño"

l. 356 and elsewhere: "La Nina" → "La Niña"

l. 311: "during dry El Niño initializations there **are** cyclonic features" (missing word)

l. 319: "include stronger negative SST  anomalies"

l. 321: "Maritime Continent" (upper case)

**REFERENCES**

E. Kalnay et A. Dalcher (1987). Forecasting Forecast Skill. Monthly Weather Review, 115(2): 349–356. doi: 10.1175/1520-0493(1987)115<0349:ffs>2.0.co;2.

---

## Author Comment (AC1)

**Response to Reviewer Comments for "Subseasonal precipitation forecasts of opportunity over southwest Asia" by Melissa Leah Breeden, John Robert Albers and Andrew Hoell**

Response to Reviewer 1

**Review of manuscript WCD-2022-555 entitled "Subseasonal precipitation forecasts of opportunity over southwest Asia" by Melissa Leah Breeden, John Robert Albers and Andrew Hoell**

**OVERALL RECOMMENDATION**

Minor revision

**SUMMARY**

This study uses a statistical model to forecast precipitation at the subseasonal timescales (weeks 3-4 and 5-6) over a specific target region (southwest Asia). The statistical model is a Linear Inverse Model (LIM), and the main goal of the manuscript is to illustrate how some intrinsic characteristics of those LIM forecasts (based on signal-to-noise ratio) enable to "forecast the forecast skill" (Kalnay and Dalcher, 1987). The authors show that the LIM expected skill is a powerful tool to detect forecasts that are indeed truly skillful. Additional analysis shows that this expected skill indicator is more relevant to detect forecasts of opportunity than indices related to ENSO and the MJO. Finally, it also enables to identify another physical signal, namely OLR anomalies in the SPCZ, that is more closely related to these forecasts of opportunity.

**MAJOR COMMENTS**

This study builds on the LIM methodology, that has already been used by some co- authors of the current manuscript to identify subseasonal forecasts of opportunity (e.g Albers and Newman 2019). To my mind, the major advances that it proposes relative to previous LIM studies are:

i) the forecast of a local meteorological parameter, i.e precipitation. This makes this study very relevant on the road to real-time and user-oriented applications of subseasonal forecasts
ii) the comparison with usual climate indices (ENSO and MJO)

iii) the identification of the SPCZ OLR anomalies as an additional physical indicator of forecasts of opportunity
Moreover, the manuscript is clearly written and provides interesting discussions on the "forecast of opportunity" concept. I think it will be fit for publication once the authors have addressed the minor revisions and my main concern below.

We and appreciate the suggested revisions listed below and thank the reviewer for their careful attention to their manuscript. We are confident that by addressing these comments, the manuscript is greatly improved and ready for publication.

My main concern is about the description of the LIM forecast setup, which I feel is rather incomplete:
1) What is the frequency of initialization during the 1 January – 20 March period? Do you start a LIM forecast everyday? Every week? Thank you for asking for clarification, yes we initialize forecasts every day during this period. We have modified the text to include 'daily' (line 139).

2) I do not completely understand what the output of your LIM forecast is exactly. My assumption would be that it provides weekly values because it is fitted on weekly averages, but I am not certain. Then, where do the biweekly forecasts come from?

For instance, do you add the weekly forecast at day 15 and 22 in order to get week 3- 4 precipitation?

Thank you for raising this point. Similar to output from numerical forecast models, for each initialization, the LIM generates forecasts of precipitation anomalies - and anomalies for all variables in the state vector – by propagating the initial conditions x(0) forward in time. For this LIM, forecasts are generated, with a daily timestep, out to a lead time of 42 days. Forecast anomalies can be averaged for a given range of lead times (ie, days 8-14 for week 2) and compared to the corresponding verification precipitation anomalies from the CHIRPS dataset to determine ACC and PCC.

As the LIM is trained on daily data that has a 7-day running mean applied to it, to isolate lower-frequency variations that still exhibit small day-to-day changes. As a result, the forecasts are similarly providing a daily forecast of the 7-day smoothed anomalies. This means that from one day's LIM forecast to another, there are changes, but they are slowly evolving because they are developed from 7-day running mean anomalies and are generated using only the EOF-truncated precipitation anomalies. We have modified the text to include more information about how the LIM forecasts are generated and what they represent:

'Similar to output from numerical forecast models, for each initialization, the LIM generates forecasts of the state vector, $\hat{\mathbf{x}}(\tau)$ (Eq. 4), by propagating the initial conditions $\mathbf{x}(0)$ forward in time. In the LIM, the propagator $\mathbf{G}(\tau)$ is determined from $\mathbf{L}$ by solving the homogeneous component of Eq. (2) (Penland and Sardeshmukh 1995). In this study, forecasts are generated at a daily timestep out to a lead time of 42 days. We note that because the LIM is trained on daily anomalies with a 7-day running mean applied, that the forecasts are also lower-frequency in nature.'

Regarding your question about weeks 3-4 and 5-6 forecasts, we average the forecasts made for lead times 15-28 and 29-42 days. This is explained in the text as outlined below in response to Point 4).

3) Why do you mix up verification on weekly (Figures 1 and 4) and biweekly values? I would have expected all results to be shown on biweekly values. We appreciate this point – we wanted to balance providing some information about each individual week's skill through the PDFs of PCC, since particularly at week 2 there is a more deterministic nature– meaning a narrower distribution of PCC - in the PDFs than the longer lead time forecasts, which offers a nice example of how skill evolves with lead time. However, we felt that there would be too much redundant analysis to show each week's ACC and relative risk separately, so we opted for considering the weeks 3-4 and 5-6 forecasts to summarize the weeks 3-6 forecast period.

4) By the way, you should be more explicit about the convention you use for week 2, week 3, etc. Thank you for this suggestion, we have included the following description at the end of section 2.2, lines 151-154:

*'Week two forecasts are the average forecast using forecast lead times 1-7 days, week two forecasts lead times 8-14 days, week three forecasts lead times 15-21 days, week four forecasts 22-28 days, week five forecasts 29-35 days, and week six forecasts 36-42 days. Weeks 3-4 and 5-6 forecasts are determined using the corresponding 14-day averaged forecasts at the corresponding lead times.'*

All in all, I think some figure illustrating the outline of the LIM forecasting process for a specific example would be welcome. We appreciate this suggestion, and hope that the added text explaining the details of the LIM has been helpful in better explaining the LIM forecast process. As noted, the LIM generates a forecast evolved from initial conditions, just as a numerical model does. We then select given lead times in the LIM forecast evolution to consider the skill of the forecasts. We have included a schematic (now Fig. 1, included below) for an example how the LIM generates a 15-day forecast.

Example: LIM forecast with lead time
$$\tau = 15 \text{ days}$$

$$\mathbf{x}(0) \longrightarrow \hat{\mathbf{x}}(15) = \mathbf{x}(0)\mathbf{G}(15)$$

State Vector          Forecast of
Initial              State Vector
conditions

Forecast Propagator: $\mathbf{G}(15) = \exp(\mathbf{L} * 15)$

Figure: Schematic of LIM forecast for a lead time of 15 days.

**MINOR COMMENTS**

l. 23: Could you include a few words to help the reader visualize what you mean by "southwest Asia" (e.g names of countries or geographical features such as the Persian Gulf, the Caspian Sea, the Himalaya)? We have added the countries included in this region to this sentence and included country labels to Fig. 3a (Formerly Fig. 2a).

Figure 1: Could you also show the distribution of PCC when considering all LIM forecasts (and not only the top or bottom 20% of expected skill)? We are able to produce this figure (below) and appreciate the suggestion, however, we are not sure that including the PDF of PCC for all forecasts offers any additional information, in this case. The purpose of this figure is to compare two groups of independent forecasts that are stratified by expected skill, to test whether the actual PCC increases when theoretical expected skill is high. Given this motivation, we opt to compare the pdfs of PCC for two independent groups. This message, we believe, does not appear further supported if we include the curve for the 'all forecasts' group (black line), because this is not an independent group of forecasts from the groups of the other two curves - which together compose 40% of all forecasts, and are therefore contributing to a large portion of the distribution in the 'all dates' group.

[Figure]

Figure: adapted former Fig. 1 (now Fig. 2): PDFs of actual skill measured by pattern correlation (PCC), for the top 20% (blue) and bottom 20% (red) of expected skill dates, as well as for all forecasts (black) for weeks of lead times a) two – d) five. The green circles represent the bootstrapped median values, and the black circles represent that bootstrapped 95[th] percentile values.

You could explicitly specify in the introduction that no numerical subseasonal forecast (e.g S2S, SubX) is used in this study, contrary to other previous studies such as Albers and Newman (2019). Thank you for this suggestion, we have included this statement (lines 78-80): 'This study will focus on LIM SFOs and does not evaluate the skill of other models, though past research suggests that forecasts generated elsewise will similarly be more skillful during LIM-identified SFOs (Albers and Newman 2019, 2020).

**TYPOS**

l.299-300: "particularly – intuitively – during the strongest events"
There seems to be one adverb too many, at least for the clarity of the sentence. I suggest rephrasing. Thank you, we have removed the word 'intuitively'
l. 300-302: "While during either ENSO phases, (...) on the mean jet and circulation". I do not understand this sentence, which seems too long. Isn't there a missing verb? I suggest rephrasing. Thank you for raising this point, we have split the sentence into two sentences as follows: 'During either ENSO phase, periods of anomalously high and low precipitation can occur due to transient, synoptic-scale disturbances forming along the subtropical jet. However, the manner in which precipitation anomalies develop differs between El Niño and La Niña, due to ENSO's influence on the mean jet and baroclinic waves (Shapiro et al. 2001; Henderson et al. 2020).'
l. 306 and elsewhere: "El Nino" → "El Niño" Thank you, we have checked the manuscript for all instances and changed them to El Niño.
l. 356 and elsewhere: "La Nina" → "La Niña" Thank you, we have checked the manuscript for all instances and changed them to La Niña.
l. 311: "during dry El Niño initializations there **are** cyclonic features" (missing word) l. 319: "include stronger

negative SST anoamlies anomalies" Thank you, we have made these corrections!

l. 321: "Maritime Continent" (upper case) We have made this change.

**REFERENCES**

E. Kalnay et A. Dalcher (1987). Forecasting Forecast Skill. Monthly Weather Review, 115(2): 349–356. doi: 10.1175/1520-0493(1987)115<0349:ffs>2.0.co;2. We have included this reference in our introduction.

Response to Reviewer 2

Subseasonal precipitation forecasts of opportunity over southwest Asia

By Melissa Leah Breeden et al.

General comments

The authors investigate sub-seasonal forecasts of opportunity for precipitation over southwest Asia using expected forecast skill from a Linear Inverse Model, in addition to assessing forecast skill at lead times beyond two weeks associated with potential sources of predictability, such as ENSO and the MJO.

The text is well written and discussed, demonstrating the contribution of this work to previous research and comparing its results with other authors. The applied methodology is clear and well-founded. This reviewer's opinion is favourable to the publication of this article, and only minor revisions are requested.

We thank the reviewer for their careful attention to our manuscript and are confident that after making the revisions as suggested, that the manuscript is much improved and ready for publication.

Specific comments

L19. South Pacific Convergence Zone (SPCZ) Thank you, we have made this change.

L57. The European Centre for Medium-Range Weather Forecasts (ECMWF) - Integrated Forecasting System (IFS) We have made this change.

L59. North Atlantic Oscillation (NAO) We have made this change.

L78-L79. 2mT, OLR, and SST already defined in L64 Thank you, we have changed the text to only include the acronyms.

L80. Climate Hazards InfraRed Precipitation with Stations (CHIRPS; Funk et al. 2015). We have modified the text.

L82-L83. Suggest including "(Trenberth, 1997; Trenberth and Stepaniak, 2001)" when defining the Niño 3.4 index. Thank you, we have included these important references.

Trenberth, K.E. (1997) The definition of El Niño. Bulletin of the American Meteorological Society 78(12):2771–2778. https://doi.org/10.1175/1520-0477(1997)078<2771:TDOENO>2.0.CO;2

Trenberth, K.E. and Stepaniak, D.P. (2001) Indices of El Niño evolution. J. Clim. 14(8):1697–1701. https://doi.org/10.1175/1520-0442(2001)014<1697:LIOENO>2.0.CO;2

L283-L284. 18 days is also the timescale when the full atmospheric response to tropical diabatic heating anomalies is seen (Jin and Hoskins (1995)). Suggest including such information. Thank you, we have included this reference

and statement: "An 18-day lag is also consistent with the circulation response to tropical diabatic heating anomalies discussed in Jin and Hoskins (1995), which peaked about 15 days after the heating occurred."

Jin and Hoskins (1995). The Direct Response to Tropical Heating in a Baroclinic Atmosphere, https://doi.org/10.1175/1520-0469(1995)052<0307:TDRTTH>2.0.CO;2

L290-L293. Wonder whether using the zonally asymmetric stream function component (i.e., zonal mean removed) can better represent the atmospheric circulation response to tropical diabatic heating anomalies. Suggest replicating such an evaluation using stream function anomalies without its zonal mean (same as for section 3.2.2). Thank you for this suggestion. We have computed the composites in Section 3.2.1 and 3.2.2 using stream function anomalies with the zonal mean removed (comparison shown below for former Figure 11). To show the full patterns, these panels do not show the significance testing done in the manuscript. We do find some modification of the El Niño wave train signature when the zonal mean streamfunction is removed. However, over southwest Asia (and subtropical north Pacific and north Atlantic) positive streamfunction anomalies are actually reduced, particularly for the La Niña initializations. Given this reduction in a key part of the anticyclone located over central southwest Asia that is suppressing rainfall, we opt not to show the zonally asymmetric streamfunction anomalies and will keep our original composites.

[Figure]

Zonally asymmetric streamfunction          Zonally symmetric + asymmetric streamfunction

Figure: Left: composite using zonally asymmetric streamfunction. Right: composite using zonally symmetric and asymmetric streamfunction (same anomalies in former Fig. 11).

L293-L296. A dry linear baroclinic model could be a useful tool to assess the contribution of tropical heating anomalies over the Indian Ocean and West Pacific in modulating the atmospheric circulation response to southwest Asia (suggest including such an evaluation as potential future assessments). Thank you for this suggestion, Barlow et al. 2002 in fact used a linear model to consider the circulation response to a diabatic heating anomaly over the West Pacific, but the focus was interannual variability. We have included a suggestion to revisit this idea on subseasonal timescales to the conclusions: 'Dry baroclinic modelling experiments with idealized heating could be used to quantify the contribution of tropical heating over the Indian Ocean and West Pacific in affecting the circulation over southwest Asia on subseasonal timescales.'

I am also interested in seeing the global regression pattern between precipitation anomalies in southwest Asia and OLR/stream function anomalies considering the entire period. This would provide the overall lead/lag observed relationships, supporting the composite results. Thank you for this suggestion, we have computed the suggested regression and indeed OLR anomalies over the SPCZ region are observed:

[Figure]

**JFM $\psi_{200}$ & OLR, Regr. w/ Precip**

We have added this figure to the supplementary material, and thank the reviewer for the suggestion. We have modified the text in Section 3.2.3: 'This linearity is further supported with the 18-day lagged regression of the southwest Asian precipitation time series with OLR (Fig. S2), although the regression pattern OLR anomalies are weaker than the composite OLR during the SFOs considered in Fig. 10 and 12.'

Fig. 9. Hard to see the SST contours (L623. Add "c" after red/blue- thank you, we have made this correction).

L319. Change "anoamlies" with "anomalies" Thank you, we have made this change

L340. Suggest including additional information here, such as: Moreover, using a dry linear baroclinic model provides the reader with a deep understanding of the role played by the basic state and thermal forcing in producing the circulation anomalies Thank you, we have added this statement "Dry baroclinic modelling experiments could be useful in disentangling the role of the basic state and thermal forcing in producing this response."

---

## Author Response (AR2)

**Response to Second Revisions for WCD-2022-555**

**Reviewer 1**

The authors have satisfactorily addressed my minor comments, leading to improved clarity of the manuscript. In my opinion the article is fit for publication.
There is just one small error to correct:
l.155 "Week two forecasts are the average forecast using forecast lead times 1-7 days" -> "Week one forecasts are the average forecast using forecast lead times 1-7 days" Thank you, we have made this correction to state 'week one'.

We thank the reviewer for their careful attention to our manuscript and have made the listed correction.

**Reviewer 2**

Dear Authors,

Before I recommend the manuscript for publication, please, consider including the technical corrections described below.
We thank the reviewer for their careful attention to our manuscript and have made the noted changes and corrections, detailed below.

====================================================

L29 - Please change "de Andrade 2018" with "de Andrade et al. 2018"
Thank you, we have made this correction.

L48 - Change "upper level" with "upper-level". We have made this change

L60 - Remove "The" We have made this correction

L92 - Change "south" with "South" Thank you, we have made this change.

L155 - Change "Week two" with "Week one" We have made this correction

Figure 3 (L617) - Do you mean "1982-2020" or "1981-2020"? We mean 1982. We noticed that earlier in the text there is a 1981 in place of 1982 when describing the data. We have made this correction (line 152).

L243 - Please revise the following sentence: "median PCC shifts statistically significant shifts". We have revised the sentence to: "For lead times between weeks 2-4, statistically significant median PCC shifts reflect the increase in skill, as do the increased probability density of forecasts with PCC > 0.5."

Supplement: Add "central" in the main title. Thank you, we have made this change

Figure S1 - "top 80%" or "top 20%"? top 20%, we have made this correction, thank you!

L281 - Do you mean "1993" or "1994"? We meant 1993, thank you.

L283 - Do you mean "2003" or "2002"? Here we mean 2002, and the low expected skill dates in the vertical gray bars.

Figure 10 - Please add information regarding black and gray contours (i.e., positive and negative streamfunction anomalies). We have added the following information: The black contours show

positive (anticyclonic) streamfunction anomalies and the gray contours show negative (cyclonic) streamfunction, contoured at an interval of $3*10^6$ m$^2$ s$^{-1}$ beginning at +/- $3*10^6$ m$^2$ s$^{-1}$.

Figure 11 (L668) - Suggest removing "are" before "shown" We have made this change.

L335 - Maritime Continent We have made this change

L351 - (2005) We have added () around 2005.

L368 - Suggest removing "are" before "therefore". Thank you, we have made this change.